# The Older the Better: Infanticide Is Age-Related for Both Victims and Perpetrators in Captive Long-Tailed Macaques

**DOI:** 10.3390/biology11071008

**Published:** 2022-07-04

**Authors:** Karlijn Gielen, Annet L. Louwerse, Elisabeth H. M. Sterck

**Affiliations:** 1Animal Behaviour and Cognition, Department of Biology, Utrecht University, 3508 TB Utrecht, The Netherlands; e.h.m.sterck@uu.nl; 2Biomedical Primate Research Centre, 2288 GJ Rijswijk, The Netherlands; louwerse@bprc.nl

**Keywords:** infanticide, social housing, sexual selection, aggression, captivity, management, husbandry

## Abstract

**Simple Summary:**

In captive primates, new males have to be introduced regularly to prevent inbreeding. Sometimes, these males commit infanticide, i.e., the killing of young infants. More knowledge of the risk factors that are associated with infanticide may lower the incidence of infanticide during male introductions. We used explanations of infanticide from wild data and the anti-infanticidal strategies of females to predict these risk factors. Next, we tested these factors using demographic data collected on captive long-tailed macaques over a long period. The ages of both infants and new alpha males are important: infants under the age of 215 days are at risk of being killed, and typically young males (≤8 years of age) commit infanticide. Therefore, to lower the risk of infanticide during male introductions in captivity, we advise introducing only males in their prime age (≥9 years of age), preferably in periods with no infants younger than 215 days of age.

**Abstract:**

In wild primates, infanticide is a risk that is especially prevalent when a new male takes over the alpha position. Insight into risk factors related to infanticide may decrease the incidence of infanticide in captivity during male introductions. We investigated several risk factors of infanticide derived from hypotheses explaining infanticide in the wild and tested this in captive long-tailed macaques (*Macaca fascicularis*) using demographic data spanning a 25.5-year period. Factors that are related to infanticide in the wild explained a large proportion, but not all incidences, of infanticide in captivity. Consistent with the wild data, infants young enough to decrease the interbirth interval (<215 days) were at risk of being killed. In contrast to studies from the wild, infanticidal males were more than 2.5 years younger than non-infanticidal males. This indicates that captive settings can lead to new risks since relatively young males may gain the alpha position, promoting infanticide. Therefore, we propose the adolescent male risk hypothesis as a captive risk factor in which subadult males pose a risk of infanticide. In conclusion, the ages of both males and infants are related to infanticide in captivity and have to be taken into account during male introductions.

## 1. Introduction

In captive groups of primates, new males have to be introduced regularly to prevent inbreeding. These introductions can be risky since they are often accompanied by an increase in aggression [1,2,3] that may lead to injuries. Moreover, new males sometimes commit infanticide, a phenomenon in which young infants are killed by a conspecific. Male infanticide in wild populations has been described in many species of primates [4,5,6]. It is typically observed when an immigrant male enters the group and takes over the alpha position or when a low-ranking resident male takes over the alpha position [7,8]. Yet, not all males commit infanticide [7,9,10,11], and not all young infants are killed during an alpha male take-over [10,12]. Male benefits and female counterstrategies to infanticide may account for these differences. These male and female strategies can be used to predict circumstances that are related to the risk of infanticide. Since infanticide is an undesirable phenomenon for zoos and primate facilities, more insight into these natural strategies may help to decrease the incidence of infanticide in captivity during male introductions.

Several hypotheses have been put forward to explain male infanticide, e.g., [4,5,6,13]. The first hypothesis is the sexual selection theory [4,14], which is widely accepted and supported [4,6,15,16,17]. This hypothesis states that by killing the unrelated offspring of a lactating female, a male reduces the time to the next ovulation, after which he can sire her subsequent offspring and thus gain a reproductive advantage. Three conditions have to be met for this hypothesis to be confirmed [14]. First, killing the infant reduces the interbirth interval (IBI) of the mother, which is suggested to be at an age younger than weaning age [7]. Second, the killed infant is unrelated to the male. It is assumed that males use a rule of thumb to appraise paternity by evaluating whether and how often he mated with the mother before conception [18]. Third, the male has to father the mother’s next offspring. Many described cases of infanticide in wild primates meet these three criteria and can therefore be attributed to the sexual selection hypothesis, e.g., [4,5,8,11,15,17,19]. From the sexual selection hypothesis, the following predictions can be made for infanticide risk in captivity: prediction (1) infants are at risk of infanticide at an age when killing them shortens the IBI of their mother; prediction (2) newly introduced immigrant males have a higher chance of being infanticidal than resident males, since resident males may be related to the infants; prediction (3) both male and female infants are equally likely to be killed since both sexes give the male a reproductive advantage when they are killed.

Second, the by-product of male aggression hypothesis [20,21] argues that infanticide is a side-effect of male–-male or male–female aggression. This hypothesis states that aggression is aimed at the mother and killing the infant is an accidental by-product. It is considered to be non-adaptive and is highly debated by some authors [8,13,22]. Yet, in our opinion, the sexual selection hypothesis and the by-product of aggression hypothesis are not completely mutually exclusive, as also suggested by Brasington et al. [7]. The by-product of aggression hypothesis is a proximate explanation. In contrast, the sexual selection hypothesis is ultimate and explains why this behavior continues to exist: it is adaptive when males target young infants. The by-product of aggression hypothesis suggests that the incidence of infanticide should correlate with the incidence of aggression during an alpha position take-over. The amount of aggression may differ between new alpha males because of individual male characteristics or how the group reacts to the new male and whether this triggers counter-aggression. A recent study [3] showed that the introductions of young males were less successful compared to the introductions of males in their prime age because of severe rates of aggression (female–male and male–female). Therefore, the following prediction for infanticide in captivity can be derived: prediction (4) when young males take over the alpha position, high rates of aggression lead to a higher incidence of infanticide compared to males in their prime age.

While males may increase their fitness by being infanticidal, for females, male infanticide is costly. Thus, to protect their infants against infanticide, females have evolved several counterstrategies [4,13]. First, females may protect their young by avoiding close contact with infanticidal males [23] or by actively defending the infant from an attacking male [15,24]. In addition, females have been observed forming aggressive coalitions against infanticidal males [15,25,26]. In captivity, we can therefore hypothesize as follows: prediction (5) female experience may influence the risk of her infant being killed, in that young and inexperienced females have a higher incidence of their infants being killed than older and experienced females. Moreover, high-ranking females may gain more female support against infanticidal males [27] as females attempt to form coalitions with higher raking females [28,29]. Thus, we propose: prediction (6) high-ranking females have a lower risk of their infants experiencing infanticide than low-ranking females.

In addition, the promiscuous mating of females in multi-male groups is thought to be a counterstrategy to infanticide [4,6,13,18,30]. By mating with several males, females can achieve paternity confusion since all resident males may be potential fathers. This has two advantages for the females. First, when there is an intra-group take-over of the alpha position, the new alpha male may be the father of the infant, and an infanticidal attack would decrease his fitness. This may prevent infanticidal attacks. Second, when an immigrant male takes over the alpha position, resident males may protect the infants because they are potentially their father. Indeed, studies have described successful resident male defense of young infants during take-overs by new males [18,24,31]. Moreover, the incidence of infanticide is lower in multi-male groups compared to one-male groups [32,33]. Promiscuous mating as a female counterstrategy to infanticide yields the following predictions concerning infanticide risk factors in captivity: similar to the sexual selection hypothesis (prediction 2), immigrant males have a higher chance of being infanticidal than resident males; prediction (7) infants in groups with more resident males may have a lower risk of being killed compared to groups with no or few resident males.

Moreover, these female counterstrategies and protection by resident males may pose a risk for new alpha males who attack infants because it can lead to lethal injuries. Therefore, infanticidal attacks are only adaptive for a male when the benefits outweigh the cost. This risk analysis may differ between individual males [34]; the ages and ranks of immigrant and resident males interact with the adaptive value of an infanticidal attack. In this model study [34], killing young infants would be more beneficial for young males in primate species with a clear reproductive skew. This leads to the same hypothesis as prediction (4), i.e., that particularly young males have a higher risk of being infanticidal than older males.

As male benefits and female counterstrategies may relate to infanticide risk, the group composition can reflect these factors to varying degrees. Subsequently, new alpha males may adopt different strategies according to the opportunities the group composition offers them [10]. For instance, many young infants in a group may increase the risk that a new male behaves infanticidally since he has more opportunities to kill infants, giving him a higher reproductive advantage. In addition, many resident males in a group may decrease the risk of a new alpha male behaving infanticidally since the risk of being wounded during infanticidal attacks increases. In particular, old, high-ranking males, who probably sired most of the infants, have been observed protecting infants successfully from infanticidal attacks [24]. When we take group composition into account, we can make the following predictions in captivity: prediction (8) groups with many young infants are at higher risk of infanticidal attacks compared to groups with few young infants; prediction (9) groups with many resident males have a lower incidence of infanticide compared to groups with few resident males; prediction (10) a higher ratio of resident males/infants at risk reduces the incidence of infanticide; prediction (11) groups with former alpha males present have a lower risk of infanticide compared to groups in which former alpha males were removed.

Overall, male benefits and female counterstrategies thus predict that infant characteristics (age and sex), male characteristics (origin and age), female characteristics (age and rank), and group composition (number of resident males, presence of the former alpha male, and number of infants) may be associated with the risk and incidence of infanticide in captivity. Since infanticide is an undesirable phenomenon for zoos and primate facilities, it is important to know which of these factors are actually related to the incidence of infanticide in captivity. Therefore, we address this issue in a long-term study of a captive colony of long-tailed macaques (*Macaca fascicularis*).

Long-tailed macaques are non-seasonal breeders that live in multi-male/multi-female groups that consist of up to 50 individuals in the wild [35]. Although paternity is skewed towards the highest-ranking male [12,36], females have been observed mating promiscuously (wild: [36,37]; captive: [38,39,40]). Females reside in their natal group that consists of several matrilines, while males disperse at the age of 4–6 years [41,42]. Often, these males enter a new group unobtrusively and may also live semi-solitary for a short period. When they reach prime age, males may attempt to obtain the alpha position. Males that fail to obtain the alpha position may either immigrate into a new group where they compete for the alpha position immediately or remain in their resident group at a lower dominance rank [41,42]. The incidence of infanticide in long-tailed macaques is quite low compared to other primate species [6], although several cases of infanticide have been reported in the wild [12,43,44] and in captivity [45,46]. These characteristics make this species suitable for studying which factors are related to infanticide. We use demographic data collected over a 25.5-year period from the captive colony of long-tailed macaques at the Utrecht University, the Netherlands.

We investigated which infant, male, female, and group characteristics were associated with the incidence of infanticide during alpha male take-overs. Unfortunately, no data on male paternity were available.

## 2. Materials and Methods

### 2.1. Study Colony

The colony of long-tailed macaques was housed at the ‘Ethologiestation’ at Utrecht University, Utrecht, the Netherlands. Our colony housed multigenerational naturalistic groups that mimicked wild demographic processes, encompassing multigenerational groups of resident females and offspring; males could change groups. All individuals were reared by their mothers in a social group.

The colony was started in 1964 with two founding groups, the Tante- and the Doka-group, and in 1980 a third group, the Burgers-group, was added. The vast majority of individuals descended from these three initial groups. Three breeding males were obtained elsewhere. The breeding of females was prolific, and group sizes increased. When a group became too large, as indicated by either the number of individuals or the level of aggression, it was typically split along genealogical lines, keeping matrilineal family groups intact as much as possible. During the study period, groups were split on a number of occasions. In 1984, at the start of the study period, the three founding groups were present. At the end, in July 2009, we housed five mixed-sex groups and two bachelor male groups. We intervened as little as possible with the behavior and group composition of the animals.

For our study, we used age classes defined by Napier and Napier (1967), as cited by [47]: (1) infants: 0–1.5 years of age; (2) juveniles: 1.5–3 years of age; (3) subadult males: 4–8 years of age, when their canines have started to grow, but their body size is not yet fully mature; (4) adult males: 9 years of age and older, when they are fully grown. For adult females, we used an age of 4 years of age and older, when they usually gave birth to their first infant in our colony.

Our mixed-sex groups consisted of single-male and multi-male groups. All groups contained at least one adult male, except in the case of the Roza-group in the period in which Flamingo was the alpha male. During an eight-month period, no (sub)adult males were present in the group in connection with an experimental study on cognition in aging monkeys [48]. A multi-male group contained an alpha male and at least one additional adult or subadult male.

*Resident males* were defined as having been resident in a specific group for at least a half year, whereas *immigrant males* were unfamiliar to a specific group. Males were introduced to groups in two situations. When a group was split, one of the new groups remained with the resident alpha male, whereas a new male was introduced to the other new group. Alternatively, when a resident male had been present for approximately four years, inbreeding with his female offspring could occur. These males were replaced as alpha male by taking them out of their social group, after which a new, unfamiliar male was introduced. When possible, these introductions took place during spring and summer, when the group had access to both their inside and outside enclosures. Because long-tailed macaques are non-seasonal breeders, it was not feasible to time introductions in periods without infants or pregnant females present in the group. Therefore, the number of infants and pregnant females in a group was not taken into account when new males had to be introduced. On several occasions, new adult males were introduced together with one or more adult or subadult males. Newly introduced males were either directly taken out of their natal group or were taken from a bachelor group. On several occasions, resident males (either natal or non-natal) rose in the dominance hierarchy and took over the alpha position. In our study, we define *a male take-over* as either a newly introduced male that reached the alpha position or a resident male that rose in rank to obtain the alpha position.

### 2.2. Data Collection, Ethics and Definitions

The data were derived from the long-term demographic records of the colony and from daily ad libitum observations of the groups. These observations were performed by experienced animal caretakers and scientific staff. All births, deaths and group changes were registered in our database; therefore, the exact group composition could be determined at any point in time. At least once a year, a genealogical tree was made for each social group. Moreover, the female dominance hierarchy was determined every 1–2 years for each social group. Furthermore, all groups were checked on a daily basis for injuries, illnesses, abnormalities and important dominance rank changes. These were noted down in welfare diaries. For our study, we used the data from 1 January 1984 to 1 July 2009. The housing conditions were described by Aureli et al. [49] (for the situation until June 1991) and Das et al. [50] (for the situation after June 1991, when the colony was moved to a new location). No ethical approval was required for this study, as all data were derived from procedures related to daily care. Moreover, no invasive or experimental procedures were performed to obtain our data.

Infants were considered *a victim of infanticide* when an infanticidal attack, leaving the infant dead, was observed or when an infant was found either dead or mortally wounded with canine bite wounds. As only males have canines that are able to puncture the skin and skull completely, all victims were considered to be killed by males. No juveniles were attacked or found dead with such wounds.

In the analyses, we only considered *infants at risk* of infanticide when their death could potentially increase a male’s fertilization chances, which normally would favor his reproductive success. According to the sexual selection hypothesis, such an infant should be young enough at the time that a male obtained the alpha position that its death would potentially reduce its mother’s IBI (see results) or when it was born in the period after a male obtained the alpha position but could not have been fathered by him (i.e., within one length of pregnancy: 162 days; [51]). During almost all male take-overs, young infants were present in the group or females were already pregnant. However, when Quattro took over the alpha position, there were no young infants in the group, nor were there pregnant females. Yet, Quattro later killed an infant who was sired at the start of his tenure; therefore, his take-over was also taken into account in our analyses.

### 2.3. Infant Characteristics

First, we analyzed which infant characteristics were related to infanticide. To test whether infants were killed at an age that reduced the IBI of the subsequent offspring, we first had to assess at what time the death of an infant would normally shorten the IBI. Therefore, we used two methods. In the first method we used the same procedure as described by Crockett and Sekulic, 1984 [52]. In accordance to that procedure we calculated the linear relationship between the age of an infant when it died within one year after birth with the IBI to the next infant and calculated the point of intersection with the average IBI of surviving offspring. For the linear relationship, we used either victims of infanticide or infants that died of other causes; infants that were born dead were not taken into account. In the second method, the age of weaning was estimated since shortening the period of lactation would give the male a reproductive advantage. Therefore, we used the following formula: period of lactation = average IBI after surviving offspring—(time to resume cycle until conception + length of pregnancy). The time to resume cycling until conception was assessed from the females of our colony who had their anti-conception measures removed until the conception of their next infant, which was on average 89 ± 20 days (*n* = 7, unpublished data). The length of pregnancy was considered 162 days [51]. After assessing, with both methods, the age at which killing infants would shorten the IBI, we tested whether victims of infanticide met this criterion.

Next, we analyzed whether the sex of an infant was related to infanticide risk. Therefore, we calculated the male/female ratio of infants that were at risk during alpha male take-overs and compared this with the male/female ratio of infants that were killed. Next, we only took infanticidal male take-overs into account and also estimated whether the male/female ratio differed between the infants that were at risk and the infants that were killed.

### 2.4. Male Characteristics

Second, we analyzed which male characteristics were related to infanticide. We assessed whether male origin was related to infanticide risk. Therefore, we counted how many infanticidal males were immigrants and how many were resident during the take-over and compared that to the number of immigrant and resident males that were not infanticidal. Next, we assessed whether male age was related to infanticide risk by comparing the average age of infanticidal and non-infanticidal males. As Flamingo was reintroduced after a period of absence, we used his age during the first introduction in our analysis.

### 2.5. Maternal Characteristics

Third, we analyzed which female characteristics were related to infanticide. We first analyzed whether female experience was related to infanticide risk. When a female gave birth to her first offspring (a nulliparous female), she was considered to be inexperienced, whereas females giving birth to subsequent offspring (parous females) were considered to be experienced mothers. We compared the number of experienced females and inexperienced females when assessing the differences between victims and infants at risk that were not killed. Next, we took female age as an estimate of maternal experience; we expected that older females had more maternal experience. Therefore, we compared the average age of mothers whose infants were killed with the average age of mothers whose infants were not killed at the time of a male take-over. There was an exception for Deha (see Appendix A); we took her age at the day that her infant was killed since this was three years after the take-over.

Furthermore, we also assessed whether maternal dominance rank was related to infanticide risk. Female long-tailed macaques have a long-term, stable linear dominance hierarchy in which daughters typically gain a rank position just below their mother, and younger daughters outrank their older sisters [35,53]. Although minor dominance rank changes may occur within a matriline (e.g., an adult daughter who outranks her mother), dominance rank for matrilines most often remain stable for several or many years. Major changes in the dominance rank of matrilines are accompanied by fights and cannot be missed. In our study, minor dominance rank changes could have occurred in the period between the last estimation of the group rank hierarchy and the period of the male take-over. Therefore, we used a less specific but reliable method for assessing the dominance rank of females with infants at risk [54]. The dominance rank of a female could be high, defined as a dominance rank position in the upper half of the ranking of all adult females, or low, i.e., in the lower ranking half. When an unequal number of females was present, the middle-ranking female was assigned to the low-rank group. We counted the number of high- and low-ranking mothers of victims and compared that to the number of high- and low-ranking females whose infants at risk were not killed. 

### 2.6. Group Composition

Fourth, we analyzed whether group composition was related to infanticide. We first assessed whether the number of infants at risk in a group was related to infanticide risk; we compared the average number of infants in groups with infanticidal males with the number of infants in groups with non-infanticidal males. To estimate whether the incidence of infanticide was higher when there were more infants in a group, we only considered groups with infanticidal males and correlated the number of infants at risk and the number of infanticide victims in a group. We then assessed the role of resident (sub)adult males in relation to infanticide risk; we compared the number of resident males between groups with infanticidal males and groups with non-infanticidal males. We also established whether the presence of the former alpha male reduced infanticide risk by assessing in how many instances the alpha male was present in groups with infanticidal males compared to their presence in groups with non-infanticidal males. Finally, we calculated the ratio of resident (sub)adult males per infant at risk and examined whether this was related to the incidence of infanticide. We first assessed whether there was a difference in this ratio between groups with an infanticidal male compared to groups with a non-infanticidal male. Next, we established, in groups with infanticidal males, whether a higher ratio of resident males per infant at risk would lead to a lower incidence of infanticide.

### 2.7. Statistics

We analyzed our data in IBM SPSS Statistics 27. We used a chi-square test to compare the sex ratio of infants that were killed with the sex ratio of infants that were at risk and also to compare the experience and rank of mothers whose infants were killed with mothers whose infants were not killed. The effect of male origin and the presence of the former alpha male were assessed by a Fisher exact test. For group comparisons, data were first analyzed for normal distribution by a Kolmogorov–Smirnov test. Then, according to the results, a t-test or Mann–Whitney U test was used. This analysis was used to compare: (1) the male age of infanticidal and non-infanticidal males, (2) the number of infants at risk in groups with infanticidal males and groups with non-infanticidal males, (3) the number of resident males in groups with infanticidal males and groups with non-infanticidal males, and (4) the ratio of resident males per infant in groups with infanticidal males and groups with non-infanticidal males. Spearman’s rho correlation was used to estimate a relationship between the number of infants in a group and the ratio of resident males per infant with the incidence of infanticide. All our statistics were tested two-tailed with a *p*-value of 0.05.

## 3. Results

### 3.1. Causes of Infant Death

We monitored the number of births and immature deaths at our captive colony of long-tailed macaques for 25.5 years, from January 1984 to July 2009. In this period, a total of 616 infants were born. Of these infants, 130 died in the first 1.5 years of their life (21%). Many of these infants were born dead (88 out of 130: 68%).

An infant was considered a victim of infanticide when the infanticidal attack was observed (*n* = 5) or when it was found dead or mortally wounded with canine bite wounds (*n* = 10). Fifteen infants (2% of all infants born or 12% of all live-born immatures that died younger than two years of age) certainly or most likely died of infanticide.

### 3.2. Infant Characteristics

We examined whether infants that were killed by infanticidal males in our population had characteristics that made them more vulnerable to infanticide (Appendix A). We tested whether infants that were killed were young enough to shorten the interbirth interval (IBI) of their mothers (prediction 1). In the first method that we used to estimate this age, the linear relationship between the age of an infant when it died with the IBI to the next infant (Figure 1) resulted in the following formula: IBI = 0.52 × (age at death) + 353. This linear relationship was not significant but showed a strong trend (*p* = 0.053). The point of intercept with the average IBI after surviving offspring (465 ± 6.8 days; *n* = 469 IBIs) resulted in a maximum age of 215 days at which infanticide would, on average, reduce the IBI of the subsequent offspring. In the second method, we estimated the weaning age through the following calculation: average IBI after surviving offspring—(time to resume cycle until conception + length of pregnancy), resulting in 465 − (89 + 162) = 214 days of age. Therefore, killing an infant under the age of 214 days would shorten the time of lactation. Since these results were almost identical, in further analyses, we used 215 days as the age under which infants in our population were at risk of being killed.

Of the infants that were killed, all 15 victims were the youngest offspring of the mother, and 13 out of 15 were younger than 215 days of age. Thus, killing these infants may have shortened the IBI.

We also tested whether male and female infants were equally likely to be killed (prediction 3). Therefore, the sex ratio of all infants at risk (162 days before birth until 215 days of age) during an alpha male take-over (M/(M + F) = 0.44, *n* = 221) was compared to the sex ratio of the infants that were killed by an infanticidal male (M/(M + F) = 0.53, *n* = 15). We found no indication of a sex bias when males killed young infants (*X*^2^(1) = 0.53, *p* > 0.05). In addition, when we only considered infants that were at risk in groups with infanticidal males (M/(M + F) = 0.49, *n* = 51), we found no evidence of a sex bias (*X*^2^(1) = 0.16, *p* > 0.05). Thus, only infant age was related to the incidence of infanticide.

### 3.3. Male Characteristics

Next, we examined which male characteristics were related to infanticide risk. During the study period of 25.5 years, 34 males took over the alpha position. Ten of these males killed, in total, 15 infants (Table 1). The majority of these males (8/10) killed infants within the first year after the start of their tenure. However, Duo killed an infant after a tenure of 3.5 years. Five months before the killing of this infant, other (sub)adult males were removed from the group, leaving Duo as the only male capable of severely wounding an infant. The other male that killed an infant after one year of tenure was Kadoo. He killed an infant 13 months after the start of his tenure. Although this infant kill was not observed, previous attempts to kill this infant had been observed. Moreover, Kadoo had already killed an infant five months after the start of his tenure. Therefore, Duo and Kadoo were considered to be the perpetrators of these infanticides.

We first assessed whether immigrant males were more often infanticidal than residents (prediction 2). Of the infanticidal males (*n* = 10), two were resident, and eight were immigrant males. Of the non-infanticidal males (*n* = 24) six were resident, and 18 were immigrant males. Therefore, no relation was found between male origin and the risk of being infanticidal (Fisher exact test, *p* > 0.05).

In addition, we expected that younger males would pose a higher risk of infanticide compared to older males (prediction 4). The age range of infanticidal males (*n* = 10) varied between 6.2 and 8.8 years of age, whereas the age range of non-infanticidal males (*n* = 24) varied between 5.5 and 14.7 years of age. Indeed, infanticidal males were significantly younger than non-infanticidal males (Mann–Whitney U; U = 39, *p* = 0.01); the age difference was approximately 2.5 years (infanticidal males: 6.9 ± 0.2 years of age versus non-infanticidal males: 9.5 ± 0.5 years of age; Figure 2). Thus, considering male characteristics, only male age was related to the incidence of infanticide.

### 3.4. Maternal Characteristics

Subsequently, we looked into the characteristics of females whose infants were at risk during an alpha male take-over. Since mothers may actively protect their infants against infanticidal attacks, we expected that experienced mothers would experience a lower incidence of infanticide compared to unexperienced mothers (prediction 5). In groups with infanticidal males, 9 out of 15 mothers of the infants that were killed had previous experience with raising young; for the mothers of the infants at risk that were not killed, 28 out of 36 had previous experience, which is not significantly different (*X*^2^(1) = 1.68, *p* > 0.05). When we considered female age, we also found no difference between the average age of mothers whose infants were killed (8.4 ± 1.2 years of age; *n* = 15) and mothers whose infants were not killed (9.6 ± 0.8 years of age; *n* = 36; *t*(49) = −1.84, *p* > 0.05).

In addition, we expected that the infants of high-ranking mothers would experience a lower incidence of infanticide than low-ranking mothers (prediction 6). However, we found no significant effect of the dominance rank of the mothers of killed infants (4 out of 15 mothers had a high rank) compared to the dominance rank of mothers of infants at risk that were not killed (15 out of 36 had a high rank; *X*^2^(1) = 1.02, *p* > 0.05). Thus, maternal characteristics did not affect the incidence of infanticide.

### 3.5. Group Composition

Lastly, to see whether group composition was related to infanticide risk (Table 1), we first determined whether the incidence of infanticide increased when there were more infants at risk in a group (prediction 8). However, no difference was found between the number of infants at risk in groups with infanticidal males (5.0 ± 1.2 infants, *n* = 10 males) and groups with non-infanticidal males (7.1 ± 0.8 infants, *n* = 24 males; *t*(32), 1.38, *p* > 0.05). Similarly, when only considering infanticidal male take-overs, we did not find a correlation between the infants at risk in a group and the number of infants that were killed (Spearman’s rho; *n* = 10, *r_s_* = 0.51, *p* > 0.05).

Next, we studied the effect of resident (sub)adult males on infanticide risk; we expected that the presence of resident males, especially the presence of the former alpha male, would lower the incidence of infanticide (predictions 9 and 11). However, we found no difference in the number of resident (sub)adult males between groups with infanticidal males (1.9 ± 0.6 (sub)adult males, *n* = 10 males) and groups with non-infanticidal males (2.8 ± 0.4 (sub)adult males, *n* = 24 males; *t*(32), 1.36, *p* > 0.05). Similarly, the presence of the former alpha male did not differ between groups with infanticidal and non-infanticidal males (Fisher’s exact test, *p* > 0.05).

Subsequently, we predicted that a higher ratio of the number of resident (sub)adult males per infant at risk would reduce the incidence of infanticide (prediction 10). Yet, we did not find a difference in this ratio between groups with infanticidal males and groups with non-infanticidal males (Mann–Whitney U; U = 109.5, *p* > 0.05). Similarly, when we only took groups with infanticidal males into account, the number of resident (sub)adult males per infant at risk was not related to the number of infants that were killed during an introduction (Spearman’s rho, *n* = 10, *r_s_* = −0.13, *p* > 0.05). Thus, group composition was not related to the risk or incidence of infanticide.

## 4. Discussion

Infanticide is a serious risk when a new male takes over the alpha position in social groups of primates. Since infanticide is an undesirable phenomenon for zoos and primate facilities, insight into infanticide risk factors in captivity may lead to management recommendations to lower the rate of infanticide. Therefore, we assessed which factors, derived from explanations of infanticide in the wild, were related to the incidence of infanticide in captive long-tailed macaques. Our most important results were that both infant age and the age of the new alpha male were related to infanticide risk. Infants younger than 215 days had a higher risk of being killed by infanticidal males, and infanticidal males were more than 2.5 years younger compared to non-infanticidal males. Other factors, such as infant sex, the characteristics of the mother and group composition, were not related to the incidence of infanticide. While several outcomes were consistent with predictions derived from the sexual selection hypothesis, proximate mechanisms specific to captivity may also partly explain the outcomes.

### 4.1. The Sexual Selection Hypothesis of Infanticide

The sexual selection hypothesis of infanticide has gained the most evidence for wild primates [5,6,15] and may also account for infanticide in captivity [16,17]. For the sexual selection hypothesis to be confirmed, the following criteria have to be met [15]: (1) killing young offspring should reduce the IBI of their mothers; (2) the victim should be unrelated to the perpetrator; (3) the perpetrator should sire the next offspring.

In line with the first criterion of the sexual selection hypothesis, males typically killed infants at an age that reduced the IBI of their mother, which was estimated at 215 days or less. However, in two cases, the infants that were killed were too old to give the new alpha males a reproductive advantage. The first infant, Pooh2, was aged 248 days, slightly older than the critical age of 215 days; however, the infant might still have triggered the proximate mechanism that males use to estimate the age of an infant. In contrast, the second older infant, Veenema, was much older at 404 days. However, it was observed that the new alpha male, Flamingo, first showed aggression against the youngest infant, Kufo (born 39 days after the take-over), who was successfully protected against these attacks by his mother Felix. Veenema was the next youngest individual in the group, so Flamingo may have diverted his infanticidal tendency towards the next youngest infant. So, in our colony of long-tailed macaques, almost all infants that were killed were young enough to give the perpetrator a reproductive advantage, making infant age a serious risk factor for infanticide. However, older infants are not guaranteed to be safe from infanticidal attacks.

Moreover, sexual selection predicts that male and female infants have an equal risk of being killed by an infanticidal male since killing both sexes gives him a reproductive advantage. Our data indeed supported the conclusion that infanticidal males have no sex bias when they kill infants. This contradicts a different hypothesis that suggests that males have a bias towards killing male offspring to eliminate future male competitors [55,56]. This bias for killing male infants has been found in chimpanzees [57], colobus monkeys [11] and spider monkeys [58]. Yet, alternative explanations can also explain the male bias found in these studies [11,59]. Moreover, many studies did not find a male bias for infanticidal males [5,8,10,16,19], which is similar to our results.

Another prerequisite for the sexual selection hypothesis is that the victims should be unrelated to the perpetrator. In our study, 5 of the 15 infants that were killed were probably fathered by the infanticidal male himself since these males had already attained the alpha position at the time of conception. Although these five cases seemingly do not fit the sexual selection hypothesis, between one and three cases may still be reconciled with the sexual selection hypothesis. One of these infants was Bapao, who was killed by Flamingo. In connection with experimental procedures [48], Flamingo was temporarily removed from the group for an eight-month period. During his first tenure, he impregnated Bapao’s mother Alfa, but he was already removed from the group when Bapao was born. One month after his reintroduction, he killed Bapao. This incidence of infanticide may still be explained by the sexual selection hypothesis because the male acted as if he killed a foreign infant. Furthermore, Nnm3 and Tequilla were sired at the start of the tenure, within one month after the male took over the alpha position. Since we do not know the exact mechanism that males use to estimate the timing of whom they have impregnated and since they also kill the infants of females who were pregnant during the take-over, a ‘miscalculation’ may account for these infant kills. However, two other infants were sired long after the male gained the alpha position. Het Loo was conceived more than 100 days after the start of the tenure, and Nnm1 was conceived more than three years after the start of the tenure, so a miscalculation cannot account for these cases. Hence, although most of the victims that were killed satisfied the second prerequisite, in captivity, males sometimes seem to kill their own offspring. Moreover, since two males killed infants more than one year after they had obtained the alpha position, infanticide in captivity is not only prevalent during the start of a tenure, as wild studies suggest [7,11,15].

In addition, following the second prerequisite, we predicted that resident males, as potential fathers, were less likely to be infanticidal compared to immigrant males that gained the alpha position. Nevertheless, we did not find evidence that resident males posed a lower risk of infanticide than immigrant males. This is in accordance with a study of wild-living white-faced capuchins [7]. Yet, we do not know whether the females whose infants were killed had mated with the infanticidal male, nor do we have paternity data for the infants. Since males may remember with whom they have mated [18] and since mating in long-tailed macaques is highly skewed towards the highest-ranking male [12,36,39], the infants that were killed were probably fathered by the former alpha male. Therefore, previously low-ranking resident males who took over the alpha position may also have gained a reproductive advantage by killing young infants, which is in line with the sexual selection hypothesis.

The third criterion of the sexual selection hypothesis, that infanticidal males should father the next offspring, was not examined in our study. When taking the first two criteria into account, a large proportion of the infant kills in our study comply with the sexual selection hypothesis. Yet, some incidences of infanticide do not fit these criteria. Hence, other mechanisms may also contribute to infanticide in captive primates.

### 4.2. Female CounterStrategies to Infanticide and Resident Male Protection

Females may employ several counterstrategies to prevent or counter infanticide [4,13]. Possibly, these counterstrategies also protect infants at risk in captive settings. First, females may protect their young against infanticide by actively defending them from infanticidal attacks [15,24] or by avoiding close contact with infanticidal males [23]. Therefore, we expected that experienced females would be more efficient in protecting their infants against infanticidal males. Yet, we did not find a systematic effect of maternal experience on the incidence of infanticide. Nevertheless, from personal observations, we saw that some females successfully defended their infants against infanticidal attacks, such as Felix, who protected Kufo against the attacks of Flamingo. Apparently, the successful protection of infants is not related to experience, so other factors may account for this, such as body weight, fighting abilities and personality.

Moreover, females may also form coalitions against infanticidal males [15,25,26]. We expected high-ranking females to gain more female support during the attacks of infanticidal males compared to low-ranking females. However, we found no evidence that female dominance rank was related to the incidence of infanticide. In our analyses, we used a rough method to determine the dominance rank of a female (high versus low); a more precise measure would have been better. Unfortunately, we did not have the precise rank order of all females during the take-over periods of new alpha males. Moreover, in nepotistic species of primates, such as long-tailed macaques, female support often comes from close kin [60], and friendship may also play a role in female coalitions [61]. Yet, in our study, the support of female kin and individuals with close social bonds against infanticidal attacks was typically not documented. Future research can focus on the role of female support from kin and friends during infanticidal male attacks.

Lastly, promiscuous mating may form an important anti-infanticidal strategy of females [4,13,18,30]. Mating with multiple males creates paternity confusion, and all potential sires may therefore protect the infants of females they have mated with against infanticidal attacks. Therefore, resident males were predicted to behave less infanticidally compared to immigrant males, and infanticide was predicted to be less frequent when there were resident males present to protect infants at risk. Yet, we found no difference in the incidence of infanticide between immigrant and resident males; similarly, the presence or number of resident males did not affect the incidence of infanticide. Unfortunately, we have no mating data regarding mothers with resident males, so it is uncertain whether the resident males were the potential sires of the infants at risk. Yet, since female mating is skewed towards the highest-ranking male [38,39] the former alpha male probably sired most infants. Indeed, studies from the wild show that former high-ranking males, in particular, successfully protected infants from being killed by infanticidal males [24]. In our colony, we typically removed the old alpha male from the group during introductions of new males; therefore, they could not protect their offspring.

Altogether, in our colony of long-tailed macaques, female counterstrategies were not related to the incidence of infanticide. The captive settings in our study may have hindered females from fully taking advantage of the counterstrategies that are proven to be successful in the wild. In addition, captive settings may present extra risks. For example, mothers with infants at risk may be unable to create a safe distance from the infanticidal male in captivity.

### 4.3. Group Composition

As not all males behave infanticidally, it is possible that they adapt their infanticidal tendency to the circumstances they are exposed to [10]. Therefore, we predicted that infanticide would be more beneficial in groups with many young infants. However, in our colony, we found no relation between the number of infants in a group and the likelihood of the new alpha male committing infanticide. Furthermore, when we only took infanticidal take-overs into account, the number of infants at risk in a group was not related to the number of victims. This is quite surprising, considering that an infanticidal male would have more opportunities to kill in groups with more infants at risk and thereby enhance his fitness. Successful protection by mothers may have prevented infanticidal males from killing more victims than would be expected based on opportunity alone.

Moreover, infanticidal attacks can be risky in groups with resident males because the perpetrator may be injured. Therefore, we expected the incidence of infanticide to be lower when there were more resident males present. In our colony, however, the number of resident males in a group did not differ between groups with infanticidal males and non-infanticidal males. Moreover, in two groups with no resident male present, no infanticide took place. In addition, when we only took infanticidal males into account, we found no relation between the number of resident males and the incidence of infanticide. Similarly, we did not find a relationship between the ratio of resident males per infant and the incidence of infanticide. This again indicates that resident males do not provide protection against infanticide.

Altogether, our data suggest that group composition does not affect whether a male behaves infanticidally. Therefore, infanticide is probably not triggered by the context a new male has been exposed to but may rather be related to characteristics of the male himself.

### 4.4. Male Characteristics

Males may have characteristics that make them more prone to commit infanticide. Indeed, we found that infanticidal males were, on average, around 2.5 years younger than non-infanticidal males. To our knowledge, we are the first study to find that male age is related to infanticide risk. Our finding corresponds to the model study of Broom et al. [34], who suggested that young males may benefit more from being infanticidal than older males. In their study, they used a simplified model that was based on the costs and benefits of an infanticidal attack in the presence of one resident male who fathered the infant at risk. Contrary to their model, our data contained many variations in the number of resident males. Additionally, most of the times when a new male was introduced in our colony, the old alpha male was removed from the group, so the likelihood that the father was present to protect the infants was very low. Therefore, it is uncertain whether, as the model proposes [34], the age difference we found can be attributed to a risk assessment based on the presence of resident males. So, other explanations have to be taken into account.

It is important to note that, in the wild, male long-tailed macaques are fully grown at 9–10 years when they obtain the alpha position [41,42], which corresponds to the average age of our non-infanticidal males. Moreover, in our colony, none of the males that were fully grown (above the age of nine) committed infanticide, whereas all infanticidal males were younger than nine years of age and not fully grown yet. Presumably, these males would not have the chance to take over the alpha position in the wild. This may explain why male age has never been described as related to infanticide before. Evolutionary explanations probably cannot account for the age difference we found since young males are unable to obtain the alpha position in the wild and therefore do not benefit from being more infanticidal. Thus, specific proximate mechanisms may explain the fact that young males pose a higher risk of infanticide in captivity. Here, we explore what these mechanisms may be.

The by-product of aggression hypothesis may account for the age difference between infanticidal and non-infanticidal males. Recently, it has been found that introductions of young male immigrant rhesus macaques are less successful due to higher levels of aggression compared to older immigrant males [3]. If infanticide is sometimes a by-product of aggression, this may also explain why young males had a higher risk of being infanticidal compared to older males in our population. Furthermore, a lack of social experience and social skills in young males may contribute to this pattern. A young male may react more anxiously and, therefore, more aggressively than older experienced males towards residents when he is introduced into a new group. Yet, our direct observations suggest that infanticidal males aim their aggression specifically at the infant and not their mother. Therefore, infant death is probably not an unfortunate by-product of male aggression against residents. Nevertheless, only 5 out of 15 cases of infanticide were witnessed; therefore, some cases of infanticide may still have been associated with high levels of aggression. Like in white-faced capuchin monkeys (*Cebus capucinus imitator*), the by-product of aggression hypothesis does not explain the overall pattern that was found for infanticide, yet some cases did meet the criteria of the by-product of aggression hypothesis [7]. Therefore, future observational research should be aimed at age-related aggression rates in relation to infanticide by new alpha males.

Alternatively, endocrine levels may explain the age difference we found. First, testosterone is associated with an increase in aggression [62]. When infanticide is a by-product of aggression, higher levels of testosterone can lead to a higher incidence of infanticide [7]. In mice, it has even been found that high levels of testosterone are directly linked to an increase in infanticide, without an increase in overall aggression [63]. If young long-tailed macaques have higher levels of testosterone compared to males at prime age, this may explain the age difference in infanticidal tendencies. To our knowledge, in long-tailed macaques, testosterone levels have only been established in juvenile and sub-adult males until the age of six and not for adult males [64]. However, in yellow baboons (*Papio cynocephalus*), testosterone peaks at the age of 6.5–7.5 years [65], which is 1–2 years before they reach full adulthood [66]. For gelada baboons (*Theropithecus gelada*), this peak was found at 6.5–8 years of age [65], whereas they are in prime age at 8.5–9 years [67,68]. These peaks of testosterone in relatively young male baboons correspond to the age of our infanticidal males, suggesting that testosterone levels may explain the age difference we have found between infanticidal and non-infanticidal males. Nevertheless, the development of testosterone levels during aging may differ between primate species [65,69], and not all primates seem to have an association between testosterone levels and the rate of aggression [70]. Research should therefore focus on testosterone development during maturation in combination with observations of aggression and infanticidal attacks in long-tailed macaques.

Moreover, in male mice, higher levels of progesterone correlate with a higher incidence of infanticide [71,72]. Theoretically, younger males may have higher levels of progesterone, which lead to a higher risk of infanticide. Little research has been performed in male primates on the levels of progesterone in relation to age; yet, in humans, there is no age effect in relation to progesterone [73]. Nevertheless, the role of male progesterone in relation to primate infanticide may be interesting to investigate further.

Altogether, male age is an important risk factor for infanticide; infanticidal males are significantly younger. Since male age has not been described as a risk factor in the wild, captive settings may thus create new risks in relation to infanticide. We propose a new hypothesis for the age difference we found in relation to infanticide in captivity: the “*adolescent male risk hypothesis*”. This hypothesis states that subadult males, in particular, may pose a risk of infanticide. Although the mechanism behind this finding is not clear yet, proximate mechanisms may explain the adolescent male risk hypothesis.

## 5. Conclusions

When we use explanations of infanticide in the wild to assess the risk factors of infanticide in captive settings, we find that some aspects in captivity are in line with the sexual selection theory. The most important finding that agrees with the sexual selection theory is that males tend to kill infants at an age that shortens the IBI of their mother. Yet, not all cases of infanticide in our colony fit the sexual selection theory. Next, female counterstrategies do not seem effective in captive settings. Although some individual mothers might have successfully protected young infants against infanticidal attacks, we found no pattern of protective factors. Presumably, the circumstances in captivity hinder females from fully exploiting these strategies. Furthermore, the infanticidal tendencies of males are probably not triggered by the circumstances of the social group but are rather related to male characteristics. We found male age to be a serious risk factor for infanticide. Since male age is not reported to be related to infanticide in the wild, captive settings may contribute to this risk factor. We propose the adolescent male risk hypothesis to describe the increased risk of subadult males being infanticidal in captivity. Future research should focus on explanations of why subadult males are more prone to commit infanticide, with attention to male aggression and the role of hormones.

## 6. Recommendations

With our results, we can give the following recommendations when new males have to be introduced in captive groups of primates:Timing of introduction. All infants whose death may shorten the inter-birth interval of their mothers are at risk of being killed during new male introductions. For captive long-tailed macaques, this age is estimated at 215 days. When new males have to be introduced, we recommend postponing the introduction until the youngest infant has reached this age. Since males also kill infants that are born after the introduction, it is also better not to introduce males when females are pregnant. Yet, sometimes postponing introductions is not advisable in captivity, for example, when inbreeding becomes a risk or when the old alpha male has suddenly died and has to be replaced immediately. In addition, especially in non-seasonal breeders, it is not always feasible to time an introduction when there are no infants at risk at all. Moreover, since we have found that infanticidal males do sometimes kill their own offspring, efficient timing cannot prevent infanticide completely. Therefore, male characteristics seem to be more practical to take into account.Male age. Since young males have a risk of being infanticidal, and older prime males do not commit infanticide, older males should be preferred when new males have to be introduced. In long-tailed macaques, males aged 9 years and older were not infanticidal, which corresponds to males that have a fully grown body size. We therefore recommend only introducing males of prime age or older. Male age may be an even more important factor for preventing infanticide than infant age.

## Figures and Tables

**Figure 1 biology-11-01008-f001:**
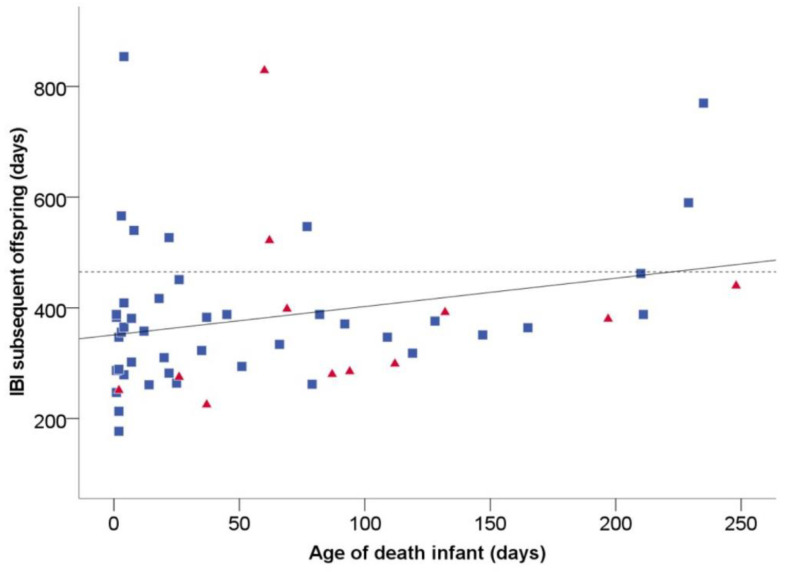
Relationship between the age of death of an infant (x-axis) and the interbirth interval to the subsequent offspring (y-axis). The scatterplot shows a linear relationship (continuous line) between the age of death of an infant and the IBI of the next offspring (IBI = 0.52 * (age at death) + 353). The horizontal dotted line represents the average IBI to the subsequent offspring after surviving offspring (465 days). The point at which the horizontal line crosses the regression line marks the age of an infant under which infanticide can decrease the IBI [52], which is 215 days. Blue blocks represent infants that died from a natural cause, and red triangles represent infants that died from infanticide.

**Figure 2 biology-11-01008-f002:**
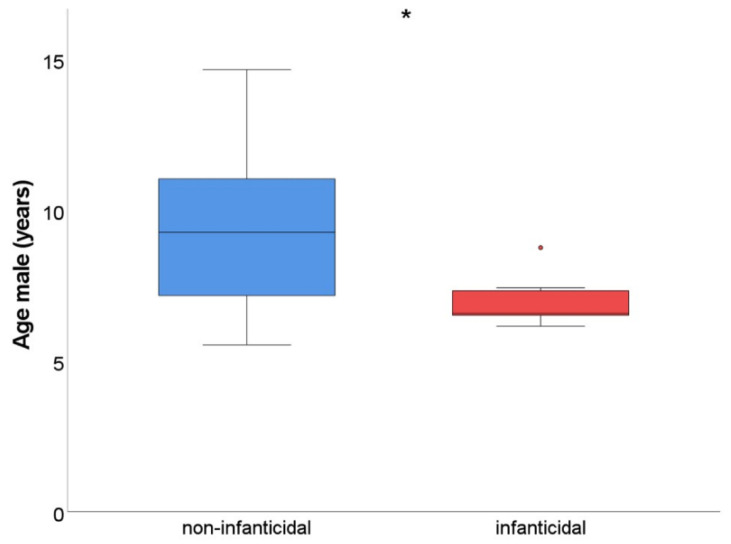
Age difference between non-infanticidal males and infanticidal males. The box plot represents the age distribution of non-infanticidal males (*n* = 24) and infanticidal males (*n* = 10). Non-infanticidal males were older compared to infanticidal males. The boxes show the age range from the first to the third quartile. The lines in the boxes represent the median, and the whiskers represent the minimum and maximum age. The one outlier in the infanticidal males was Gruo, who was 8.8 years of age. Note that this outlier is still younger than the median of the non-infanticidal male age. * *p* < 0.05.

**Table 1 biology-11-01008-t001:** Alpha male and group characteristics during new alpha male take-overs.

Type	Alpha Male Name	Origin	Age (Years)	(Sub) Adult Males (*n*)	Former Alpha Male Present	Infants at Risk (*n*)	Infants Killed (*n*)
I	Cleo	IM	6.5	0	no	1	1
I	Duo	IM	6.2	6	no	10	1
I	Ekzekwo	IM	6.5	0	no	2	1
I	Flamingo ^a^	IM	6.6	1	yes	7	2
I	Gruo	IM	8.8	1	no	9	3
I	Kadoo	RM	6.5	4	no	10	2
I	Quattro	IM	7.3	1	no	0	1
I	Ragazzo	RM	6.5	3	yes	7	1
I	Repho	IM	7.4	3	no	3	2
I	Wonpo	IM	6.7	1	no	1	1
NI	Bufo	RM	7.1	3	yes	7	0
NI	Dodo	IM	11.1	4	no	4	0
NI	Ekko	IM	9.3	1	no	1	0
NI	Gekko	IM	14.7	4	no	6	0
NI	Indy	IM	7.1	2	no	7	0
NI	Info	RM	5.5	4	no	11	0
NI	Ivo	IM	9.9	4	no	10	0
NI	Kluo	IM	11.0	0	no	6	0
NI	Marokko	IM	9.2	0	no	2	0
NI	Milo	RM	6.3	3	yes	6	0
NI	Montuur	IM	10.5	1	no	11	0
NI	Moso	IM	13.7	3	no	11	0
NI	Mucho	IM	10.5	1	no	4	0
NI	Nacho	IM	9.4	2	no	1	0
NI	Polo	IM	7.2	3	no	8	0
NI	Quayo	IM	7.1	2	yes	12	0
NI	Regilio	IM	11.2	6	no	1	0
NI	Shampoo	RM	7.2	3	yes	7	0
NI	Statusquo	IM	9.2	1	no	4	0
NI	Veto	RM	7.8	3	yes	13	0
NI	Vip	IM	12.7	7	no	15	0
NI	Voodoo	IM	14.4	3	no	10	0
NI	Xiano	RM	7.0	3	no	3	0
NI	Xom	IM	8.3	5	no	11	0

^a^ Male that killed one infant at the start of his first tenure and one infant during a reintroduction after an absence of eight months. I = infanticidal, NI = non-infanticidal, IM = immigrant male, RM = resident male. Male age was determined at the time of the alpha position take-over.

## Data Availability

Data are available on request.

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
