# Peer review of "The Older the Better: Infanticide Is Age-Related for Both Victims and Perpetrators in Captive Long-Tailed Macaques"

_biology, 2022, doi:10.3390/biology11071008_

Round 1

Reviewer 1 Report

This article is original and provide important information on infanticide in zoo primates, guiding decision-making process and management, improving macaque ex-situ conservation. I feel not qualified to edit and judge the English, but the text reads fluent and I did not report any mistake (but see some minor suggestions). My major concern are:

-        Authors use both parametric and non-parametric statistical tests based on the distribution of the datapoints. However, I am not sure this is scientifically valid, and I think only non-parametric tests should be used. The sample size is quite small so this would be a better strategy.

-        Authors analyzed the effect of dominance hierarchy/rank on the likelihood of infanticide without a reliable assessment of the hierarchy itself. I think this should be avoided. The subsample of data of infanticide males is quite small (N = 15, with only 5 direct observations of real infanticide by new alfa-males), some results are speculative. I appreciate that authors underline the limitations of their study in the discussion but this part on dominance hierarchy should be deleted.

-        If possible, authors should provide information about the rearing of the males (and females) of the study. Hand-rearing is known to impair social competence of primates, and this could affect for example the rate of aggression and the likelihood of infanticide, even by males that sired the killed infant.

-        L144: authors talk about “our” captive colony of long-tailed macaques, but this has never been mentioned before. The same happens later in the paragraph.

-        LL178-187: How did authors define age classes? Maybe some references on macaque aging are needed.

-        Who collected data during the 25-year study period?

-        From an ethical point of view, I wonder whether 1) introduction of new males could have been done when infants were old enough to prevent them to be killed/avoid infanticide; 2) females with infants at risk of infanticide could be separated and kept in sub-groups during the introduction of the new male.

-        All p-values and values from statistical tests (U, T, etc.) should be in italics.

Author Response

Point 1. Authors use both parametric and non-parametric statistical tests based on the distribution of the datapoints. However, I am not sure this is scientifically valid, and I think only non-parametric tests should be used. The sample size is quite small so this would be a better strategy.

Answer point 1: We have looked into this. Indeed, to run a Pearson test for correlation the sample size has to be at least 30. Therefore, we re-analyzed the correlations with Spearman rho. Yet, for a t-test no minimum sample size is required, and the requirements to conduct a t-test were met when they were performed. Adjustments related to the Spearman rho correlation test are made in line 335, 448, 463

Point 2. Authors analyzed the effect of dominance hierarchy/rank on the likelihood of infanticide without a reliable assessment of the hierarchy itself. I think this should be avoided. The subsample of data of infanticide males is quite small (N = 15, with only 5 direct observations of real infanticide by new alfa-males), some results are speculative. I appreciate that authors underline the limitations of their study in the discussion but this part on dominance hierarchy should be deleted.

Answer point 2: Long-tailed macaque females have a very stable linear rank hierarchy, in which daughters typically inherit the rank of their mother and younger sisters usually outrank older siblings. The exact rank order was determined every 1-2 years for each social group. However, minor rank changes could occur in the meantime. Therefore, we were not able to use the specific rank order of each individual female. Instead we used a less specific, yet still reliable method of the rank position of mothers at risk (also used by van Noordwijk, et al., 1987). In the methods we clarified why the less specific method we used is still a reliable measure for their dominance rank. In the text we added extra information and references to explain why we chose this less specific method (line 288-303).

Point 3. If possible, authors should provide information about the rearing of the males (and females) of the study. Hand-rearing is known to impair social competence of primates, and this could affect for example the rate of aggression and the likelihood of infanticide, even by males that sired the killed infant.

Answer point 3: All primates born in our colony were raised by their mother in a social group (also all males in our study). Extra information concerning this remark was added to the methods (line 169-170).

Point 4:  L144: authors talk about “our” captive colony of long-tailed macaques, but this has never been mentioned before. The same happens later in the paragraph.

Answer point 4: Valid remark. We moved this information to the methods where it makes more sense that we use ‘our’ (line 166-170).

Point 5. LL178-187: How did authors define age classes? Maybe some references on macaque aging are needed.

Answer point 5: The age-classes were redefined according to the age-classes that were used in other studies (e.g. Santosa 2012). The age classes with reference can be found in line 182-187.

Point 6. Who collected data during the 25-year study period?

Answer point 6: Data were collected by experienced animal care-takers and scientific staff. Extra information concerning this remark was added to the methods (lines 214-215)

Point 7. From an ethical point of view, I wonder whether 1) introduction of new males could have been done when infants were old enough to prevent them to be killed/avoid infanticide; 2) females with infants at risk of infanticide could be separated and kept in sub-groups during the introduction of the new male.

Answer point 7: Concerning 1). Because long-tailed macaques are non-seasonal breeders, it was not feasible to time an introduction in a period without infants or pregnant females present in the group. Therefore, the number of infants and pregnant females in a group were not taken into account when new males had to be introduced. Extra information concerning point 1) was added to the methods. (202-205) Concerning 2): Dominance rank of females is highly based on the matriline individuals belong to. When several individuals, or key individuals are removed from a matriline this may lead to huge fights when lower ranking matrilines want to take over the rank position of high ranking matrilines. Moreover, in all matrilines female bonds are very important, removing social allies may lead to stress, social instability and an increase of aggression. Therefore, following the colony management guidelines, matrilines should be held together and no females could be removed temporally from the social group. Besides, ‘temporary’ could result in almost a year if we have to separate pregnant females until their infants have reached the age of 215 days. In our opinion it would be unethical to remove females temporarily from a group.

Point 8. All p-values and values from statistical tests (UT, etc.) should be in italics.

Answer point 8: Following this remark, we adapted this throughout the paper, also we replaced comma’s in decimals for dots and added spaces between math signs.

Reviewer 2 Report

 The Older the Better: Infanticide is Age Related in Both Victims and Perpetrators in Captive Long-tailed Macaques

In the above-named manuscript, the authors analyzed long-term demographic data of a captive long-tailed macaques colony to test several factors that may affect the risk of infanticide after an alpha-male take-over; including infant -, mother -, male -, and social group characteristics.

I find the manuscript very informative. It is well-structured, the hypotheses are clearly stated and carefully derived from the literature, and the data were analyzed very thoroughly. I just have a couple general and specific comments (please see below).

GENERAL COMMENTS

#Effect of infant age on infanticide risk & Figure 1

I am not entirely sure I understood the analysis of prediction 1 (effect of infant age on the risk of infanticide) correctly. The way I understand it, the authors tested how the time of an infant’s death affects the mother’s inter-brood interval to establish an estimate of infant age at risk of getting killed. Therefore, they did a linear regression with age of dead infants as the predictor and IBI to the subsequent brood as the response. The y-axis interception of the resulting regression line with average IBI of surviving offspring then gives an estimate of which infants are at risk of being killed (Figure 1).

I am not entirely sure what the sample is. In 2.3 the authors said, the regression was done using all infants that died within one year (so that would include both infants that were killed or died for other reasons) (lines 234 - 236) (with or without the ones that were born dead?) but in section 3.2, the authors said they tested whether infants that were killed were young enough to shorten the interbirth interval (333-335). Or are these two different regressions?

In Figure 1, the authors show that, the younger a mother’s infant a died, the shorter the interval to receiving the next infant is (Figure 1). Although there is a very strong trend, I am afraid, this regression is strictly speaking not significant (p-value is 0.053 > 0.05).

Infants were defined as individuals with 0-1 years of age (line 178). But the number of infanticide victims (N = 15) is based on infants that died from infanticide within the two years of their life (lines 329-330). Please clarify.

Maybe I have missed it, so please bear with me! But I don’t see where exactly the effect of infant age on the risk of infanticide has been tested. That is, in total, 15 infants were victims of infanticide. But we don’t know, if the average age of killed infants differs from the average of not killed infants.

#Table 1

I appreciate that the authors attach their raw data but I am wondering if Table 1 wouldn’t be better placed in the supplement? I suggest, instead of imbedding the raw data in the main text, providing descripting/summarizing statistics in the main text only.

Also, and although this can be gathered from context, the abbreviations K and NK have not been explained.

Infant age is counted from the start of pregnancy onwards with day 0 being the day they were born – so an infant cannot possibly be younger than -162 days, right? I don’t understand how infants can be, e.g., -1205 days at the start of tenure.

#New alpha-male introduction

The way I understood the study, I thought, the authors aimed on testing infanticide after a new male introduction. But, e.g., Nnm1 was killed 3 years after the tenure start (line 497) (btw: I understand now how Nnm1 could be -1205 days at the start of tenure), so the study is not about new alpha-males specifically but about infanticide more generally (regardless of whether the alpha male is new or not)?

On that note, it might be worthwhile to give a more precise definition on the time frame that was considered as a new male introduction.

SPECIFIC COMMENTS

#Lines 159-161

How many groups?

#Line 167

Please add the location of the Ethologiestation of Utrecht University

#Line 174

Here and elsewhere (e.g., line 201, 178, 234), comma missing after “during the study period“.

#Line 182

I don’t understand this sentence: shouldn’t mixed-sex groups always have members of both sexes, per definition?

#Line 209

How often is regularly?

#Line 285

“We then assessed”; not “we than assessed” (i.e., “then” with and “e” not with “a”)

#Line496

Typo: “Het” instead of “Yet”

#Line 251

Delete “themselves”?

Author Response

Point 1. I am not entirely sure I understood the analysis of prediction 1 (effect of infant age on the risk of infanticide) correctly. The way I understand it, the authors tested how the time of an infant’s death affects the mother’s inter-brood interval to establish an estimate of infant age at risk of getting killed. Therefore, they did a linear regression with age of dead infants as the predictor and IBI to the subsequent brood as the response. The y-axis interception of the resulting regression line with average IBI of surviving offspring then gives an estimate of which infants are at risk of being killed (Figure 1).

Answer point 1: Your interpretation of the method is correct. This follows published and established methods to assess the infanticide risk in wild primates.

Point 2. I am not entirely sure what the sample is. In 2.3 the authors said, the regression was done using all infants that died within one year (so that would include both infants that were killed or died for other reasons) (lines 234 - 236) (with or without the ones that were born dead?) but in section 3.2, the authors said they tested whether infants that were killed were young enough to shorten the interbirth interval (333-335). Or are these two different regressions?

Answer point 2: Our objective was to test whether the victims were young enough to decrease the IBI of the subsequent offspring. Before we could test this, we first had to determine at what age killing the infants would shorten the IBI. Therefore, we made a regression of all infants that had died within their first year. To make this more clear in the methods, we added the following text: “To test whether infants were killed at an age that reduces the IBI of the subsequent offspring, we first had to assess at what time the death of an infant would normally shorten the IBI. Therefore, we used two methods.” (line 244-247). In the results we added: “In the first method that we used to estimate this age, the linear relationship between the age of an infant when it died with the IBI to the next infant (Figure 1) resulted in the following formula:….” (line 352-354).

Point 3. In Figure 1, the authors show that, the younger a mother’s infant a died, the shorter the interval to receiving the next infant is (Figure 1). Although there is a very strong trend, I am afraid, this regression is strictly speaking not significant (p-value is 0.053 > 0.05).

Answer point 3: We added a remark that the regression shows a strong trend that was not significant. (line 355)

Point 4. Infants were defined as individuals with 0-1 years of age (line 178). But the number of infanticide victims (N = 15) is based on infants that died from infanticide within the two years of their life (lines 329-330). Please clarify.

Answer point 4: According to a remark of the other reviewer, we used age-classes from literature (e.g. Santosa 2012). They described the age class of infants as 0 – 1,5 years of age, which covers the ages of the victims. We changed the age classes in the methods and also adjusted the descriptive results of the number of death to an age of 1,5. (line 182-183 & 340-343)

Point 5. Maybe I have missed it, so please bear with me! But I don’t see where exactly the effect of infant age on the risk of infanticide has been tested. That is, in total, 15 infants were victims of infanticide. But we don’t know, if the average age of killed infants differs from the average of not killed infants.

Answer point 5: We would like to give some background of the husbandry of out colony, since this may clarify the situation. The husbandry of group dynamics in our colony followed several crucial features of wild macaques: offspring remained with their mother at least until adolescence, females remained in their maternal group and juvenile / adolescent males were typically removed when they became too aggressive. Adult males were introduced about every 4-5 years to prevent inbreeding. This resulted in multigenerational naturalistic groups. Thus, both typically multiple infants and juveniles are present in a group. We recorded all attacks on and deaths of group members. We have witnessed lethal aggression of (sub)adult males on infants and have found infants dead with male bite wounds. We have never observed such attacks or deaths in juveniles. We added this information in the manuscript, line 230-231. Thus, victims of lethal male aggression were only infants.

Also, determining the average age of victims relative to infants that are not killed was not relevant for our question. Our objective was to find out if infants of an age that would shorten the IBI would have a higher risk of being killed. 13 out of 15 victims are indeed young enough to shorten the IBI, this fits with our prediction of the maximum age that an infant is at risk. It is hard to make a comparison with the average age of infants that are not killed. For victims it is clear at what age they are killed, but for infants that are not killed we cannot estimate when males would have killed them. In other words, we have no actual age to compare it with.

Point 5 (table 1). I appreciate that the authors attach their raw data but I am wondering if Table 1 wouldn’t be better placed in the supplement? I suggest, instead of imbedding the raw data in the main text, providing descripting/summarizing statistics in the main text only.

Answer point 5: Good suggestion, we put the table in the supplement (table S1).

 Point 6 (table 1). Also, and although this can be gathered from context, the abbreviations K and NK have not been explained.

Answer point 6: The abbreviations were in the subscript, but we now provided this information in the heading of the table to make it more clear.

Point 7 (table 1). Infant age is counted from the start of pregnancy onwards with day 0 being the day they were born – so an infant cannot possibly be younger than -162 days, right? I don’t understand how infants can be, e.g., -1205 days at the start of tenure.

Answer point 7: Infants younger than -162 days were probably sired by the perpetrator himself, in table S1 “b” marks infants that were younger than this age. So the infant age of -1205 means that the infant was born a long time after the male obtained the alpha position.

Point 8 (New alpha-male introduction). The way I understood the study, I thought, the authors aimed on testing infanticide after a new male introduction. But, e.g., Nnm1 was killed 3 years after the tenure start (line 497) (btw: I understand now how Nnm1 could be -1205 days at the start of tenure), so the study is not about new alpha-males specifically but about infanticide more generally (regardless of whether the alpha male is new or not)?

Answer point 8: Although infanticide has typically been described to be committed during the start of a tenure, the timing of the infant kill was not an objective of our study. Therefore, all victims of infanticide were taken into account, regardless of the timing of the infant kill. Since some formulations indeed seemed to indicate that we were primarily interested in infanticide during the start of a tenure, we changed some formulations throughout the paper, the most important change is in line 385: “…which characteristics of new alpha males were related to infanticide risk” was changed into “which male characteristics were related to infanticide risk”

 Point 9 (New alpha-male introduction). On that note, it might be worthwhile to give a more precise definition on the time frame that was considered as a new male introduction.

Answer point 9: As mentioned above, we were interested in the circumstances of all infants kills. Yet, in our opinion it is very interesting that in our study two males killed infants long after they had obtained the alpha position, since this differs from studies of wild primates. Thanks to your remark, we now added more information about the timing of a kill in the results. We mentioned that 8/10 males killed infants in their first year of tenure and we provided more information of the males that killed the infants long after they had obtained the alpha position (387-395). Also in the discussion we added a remark that in captivity the timing of infanticide may differ from wild studies. “Moreover, since two males killed infants more than one year after they had obtained the alpha position, infanticide in captivity is not only prevalent during the start of a tenure as wild studies suggest” (line 528-530).

Point 10. #Lines 159-161 How many groups?

Answer point 10: This sentence is moved to the method section (166-167). The alinea below describes the number of groups in our colony. (171-181)

 Point 11. #Line 167 Please add the location of the Ethologiestation of Utrecht University

Answer point 11: The location is added “Utrecht, the Netherlands”. (159, 166-167)

Point 12. #Line 174 Here and elsewhere (e.g., line 201, 178, 234), comma missing after “during the study period“.

Answer point 12: We have checked the document and added comma’s where they were missing.

Point 13. #Line 182 I don’t understand this sentence: shouldn’t mixed-sex groups always have members of both sexes, per definition?

Answer point 13: This sentence explains that we kept both single- and multi-male groups. Because we also had bachelor groups at the end of the study period, we chose to use ‘mixed-sex group’ in this sentence.

 Point 14: #Line 209 How often is regularly?

Answer point 14: Every 1-2 years. We added this information in the methods. (line 218-219)

Point 15. #Line 285 “We then assessed”; not “we than assessed” (i.e., “then” with and “e” not with “a”)

Answer point 15: Adjusted

Point 16. #Line496 Typo: “Het” instead of “Yet”

Answer point 16: This is not a typo, the name of the infant was actually “Het Loo”, named after a Dutch palace.

Point 17.  #Line 251 Delete “themselves”?

Answer point 17: ‘Themselves’ was deleted.

Round 2

Reviewer 1 Report

The manuscript has improved and I think it deserves publication.